

# Improvement of morphophysiological and anatomical attributes of plants under abiotic stress conditions using plant growth-promoting bacteria and safety treatments

Wasimah Buraykan Alshammari[1], Kholoud Alshammery[1], Salwa Lotfi[1], Haya Altamimi[1], Abeer Alshammari[1], Nadi Awad Al-Harbi[2], Dragana Jakovljević[3], Mona Hajed Alharbi[4], Moustapha Eid Moustapha[5], Diaa Abd El-Moneim[6] and Khaled Abdelaal[7]

[1] Department of Biology, College of Science, University of Hail, Hail, Saudi Arabia
[2] Biology Department, University College of Tayma, University of Tabuk, Tabuk, Saudi Arabia
[3] Department of Biology and Ecology, Faculty of Science, University of Kragujevac, Kragu-jevac, Serbia
[4] Department of Biology, College of Scince and Humanities in Al-Kharj, Prince Sattam bin Abdulaziz University, Al-Kharj, Saudi Arabia
[5] Department of Chemistry, College of Science and Humanities in Al-Kharj, Prince Sattam bin Abdulaziz University, Al-Kharj, Saudi Arabia
[6] Department of Plant Production (Genetic Branch), Faculty of Environmental Agricultural Sciences, Arish University, El-Arish, Egypt
[7] EPCRS Excellence Center, Plant Pathology and Biotechnology Lab, Faculty of Agriculture, Kafrelsheikh University, Kafrelsheikh, Egypt

Corresponding authors
Diaa Abd El-Moneim,
dabdelmoniem@aru.edu.eg
Khaled Abdelaal,
khaled.elhaies@gmail.com

## ABSTRACT

Drought and salinity are the major abiotic stress factors negatively affecting the morphophysiological, biochemical, and anatomical characteristics of numerous plant species worldwide. The detrimental effects of these environmental factors can be seen in leaf and stem anatomical structures including the decrease in thickness of cell walls, palisade and spongy tissue, phloem and xylem tissue. Also, the disintegration of grana staking, and an increase in the size of mitochondria were observed under salinity and drought conditions. Drought and salt stresses can significantly decrease plant height, number of leaves and branches, leaf area, fresh and dry weight, or plant relative water content (RWC%) and concentration of photosynthetic pigments. On the other hand, stress-induced lipid peroxidation and malondialdehyde (MDA) production, electrolyte leakage (EL%), and production of reactive oxygen species (ROS) can increase under salinity and drought conditions. Antioxidant defense systems such as catalase, peroxidase, glutathione reductase, ascorbic acid, and gamma-aminobutyric acid are essential components under drought and salt stresses to protect the plant organelles from oxidative damage caused by ROS. The application of safe and eco-friendly treatments is a very important strategy to overcome the adverse effects of drought and salinity on the growth characteristics and yield of plants. It is shown that treatments with plant growth-promoting bacteria (PGPB) can improve morphoanatomical characteristics under salinity and drought stress. It is also shown that yeast extract, mannitol, proline, melatonin, silicon, chitosan, α-Tocopherols (vitamin E), and biochar alleviate the negative effects of drought and salinity stresses

through the ROS scavenging resulting in the improvement of plant attributes and yield of the stressed plants. This review discusses the role of safety and eco-friendly treatments in alleviating the harmful effects of salinity and drought associated with the improvement of the anatomical, morphophysiological, and biochemical features in plants.

## INTRODUCTION

Plants face many abiotic stresses including drought (*Khaffagy et al., 2022*; *El Sabagh et al., 2019*; *Abdelaal et al., 2021a*), heat (*Soliman et al., 2018*; *Farooq et al., 2017*), and salinity (*El-Banna & Abdelaal, 2018*; *Alnusairi et al., 2021*; *Helaly et al., 2017*) that limit yield and productivity. The adverse effects of drought can be observed in morphological characteristics like leaf number and plant height in different plant species including barley (*Abdelaal et al., 2020a*), faba bean (*Abdelaal, 2015a*), pea (*Arafa et al., 2021*), and sugar beet plants (*AlKahtani et al., 2021a*). The negative effects on physiological features, such as concentrations of chlorophyll and carotenoids were observed in barley (*Hafez et al., 2020a*; *Hafez et al., 2020b*) and maize (*Abdelaal et al., 2017*). Production of reactive oxygen species (ROS) was increased significantly due to drought conditions among which hydrogen peroxide and superoxide are the main types of reactive oxygen species recorded in drought-stressed plants (*Abdelaal, Mazrou & Hafez, 2020*; *Abdelaal et al., 2022a*). The decrease in chlorophyll concentrations and relative water content was noticeable under water deficit conditions in faba bean plants (*Abdelaal, 2015a*) and heat conditions in tomato plants (*Elkelish et al., 2020*). Additionally, the decreases in the content of chlorophyll and fatty acids, as well as the photosystem II efficiency were recorded as a response mechanism to stress (*Soliman et al., 2018*). *Farooq et al. (2017)* reported that the activity of antioxidant enzymes and the photosynthesis process of chickpea plants were negatively affected under heat-stress conditions.

In addition to drought, salinity is one of the most harmful abiotic stress factors that negatively affect morphological and physiological characteristics in various economically important plants including rice plants (*Hafez et al., 2020a*; *Hafez et al., 2020b*), faba bean (*El-Flaah et al., 2021*; *Abdelaal et al., 2021b*), pea plants (*Abdelaal, Mazrou & Hafez, 2022*), and sweet pepper plants (*ALKahtani et al., 2020*). The oxidative damage can be seen in salt-stressed plants, and this is associated with the production of malondialdehyde (MDA) and ROS, particularly superoxide, and hydrogen peroxide (*Al-Shammari, Altamimi & Abdelaal, 2023*). Many studies were conducted to find suitable and safe compounds and methods to improve plant growth and yield under various stress conditions. The application of plant growth-promoting bacteria (PGPB) is among the most significant methods to overcome oxidative stress and improve plant growth (*Abdelaal, 2015b*; *Yadav et al., 2017*; *Kour et al., 2020*). *Azospirillum, Paenibacillus, Pseudomonas, Bacillus, Azotobacter,* and *Rhizobium* are the most common genera used to improve plant growth under both natural and

stressful conditions. The significant role of PGPB could be attributed to the production of compounds such as siderophores, phytohormones, and antibiotics (*Çakmakçi et al., 2006*). Inoculation of *Pseudomonas fluorescens* and *Azospirillum brasilense* caused an increase in the growth characteristics of *Urochloa* spp. (*Hungria et al., 2021*). Additionally, the application of safety compounds such as silicon, chitosan, and melatonin improved several plants' growth characteristics. *Abdelaal et al. (2021c)* reported that applying chitosan led to the improvement of morphological and physiological attributes of garlic plants under drought-stress conditions. Also, *Al-Shammari, Altamimi & Abdelaal (2023)* found an increase in seed yield and improvement in physiological features of pea plants with the application of nano silica and melatonin under salinity stress conditions. Antioxidant metabolism and the physiological traits of the salt-stressed lettuce plants were improved with the application of silicon and *Bacillus thuringiensis* (*AlKahtani et al., 2021a*). The present work aims to summarize the pivotal role of various PGPB and eco-friendly treatments as a promising strategy to alleviate drought and salinity's negative impacts and the yield and production of economically important plants under stressful conditions.

### Intended audience
The abiotic stress factors are significant for the researchers and broader audience because of the harmful effects of these factors on agricultural production and sustainable development. Our review is essential for researchers in plant breeding for abiotic stress, plant physiology, and genetics.

### Survey methodology
We have checked about 480 articles and book chapters in this field to achieve our aim and introduce the recent information and results to the researchers and audience. We used search engines like Web of Science and PubMed. The date range searched was between 1999 and 2023. We focused on the manuscripts and book chapters that summarize the pivotal role of various PGPB and eco-friendly treatments as a promising strategy to alleviate the negative impacts of drought and salinity and increase plant yield and production under stressful conditions.

## EFFECT OF DROUGHT STRESS ON MORPHOPHYSIOLOGICAL AND BIOCHEMICAL CHARACTERISTICS OF PLANTS

### Effect of drought stress on morphological characteristics of plants
The growth characteristics of plants can be significantly affected by drought stress (*Abdelaal et al., 2022b*). This is a serious threat to agricultural production (*Abdelaal et al., 2021d*) since can affect all stages of plant growth and development, including germination, seedling growth, and yield (*Elkoca et al., 2007*; *Kaya et al., 2006*). As mentioned, the reduction in leaf number, leaf area, and plant height was recorded in many plants under drought stress, including faba bean plants (*Abdelaal, 2015a*), barley (*Abdelaal et al., 2018*; *Abdelaal et al., 2020b*) and wheat (*Kutlu et al., 2021*). Similar effects of drought were recorded in sugar beet (*Abdelaal, 2015b*) and wheat (*Abdelaal et al., 2021d*). Additionally, the decrease in fresh and dry weight is a common negative effect under drought stress conditions (*Abdelaal et al.,*

**Table 1  Application of safety compounds to improve the morphological characteristics of economic plants under drought stress conditions.**

| Treatments | Improved morphological characters under drought stress conditions | Plant species | References |
|---|---|---|---|
| Proline | Increase leaves number | Sugar beet | *AlKahtani et al. (2021a)* |
| Silicon | Improve root yield | Sugar beet | *AlKahtani et al. (2021a)* |
| Proline | Increase water availability | Faba bean | *Al-Shammari, Altamimi & Abdelaal (2023)* |
| Yeast extract | Increase yield | Garlic | *Abdelaal et al. (2021c)* |
| Mannitol | Improve growth characters | Wheat | *Abebe et al. (2003)* |
| $K_2SiO_3$ | Improve shoots number, shoot length, roots number and survival percentage | Banana | *Aziz et al. (2023)* |
| Silicon | Improve fresh weight, dry weight, leaf area, and root length | Coriander | *Mahmoud et al. (2023)* |
| Biochar | Increase plant height, root dry weight and root diameter | Okra | *Jabborova et al. (2021)* |
| Biochar | Improve growth characters and yield | Tomato | *Usman et al. (2023)* |
| Chitosan | Increase plant height, number of seeds per spike, number of spikes and grain yield | Wheat | *Jahanbani et al. (2023)* |
| *Azotobacter vinelandii, + Pseudomonas putida* | Increased germination indices, dry weight, stem length, and root length | *Festuca ovina* | *Rigi, Saberi & Ebrahimi (2023)* |

*2018*) observed in many plants (*AlKahtani et al., 2021a*; *Galal et al., 2023*). Previous studies showed that water deficit caused significant decreases in morphological characteristics of faba bean and maize plants (*Abdelaal, 2015a*; *Abdelaal et al., 2017*). Moreover, a significant decrease in leaf area, plant height, and number of leaves per plant in pea was observed during two seasons under drought conditions. Also, pod number and dry weight of 100 seeds decreased significantly in drought-stressed pea plants (*Arafa et al., 2021*). In recent years many studies have been implemented to overcome the negative effects of drought stress on the morphological characters in plants (Table 1).

## Effect of PGPB and safety treatments on morphological characteristics of plants under drought conditions

The harmful effects of drought stress can be alleviated with the application of safety treatments and compounds such as PGPB, yeast extract, mannitol, melatonin, silicon, trehalose, chitosan, $\alpha$-Tocopherols (vitamin E) or biochar (Table 1). In sugar beet plants, the application of silicon and proline improved leaf number, root fresh weight, and root yield under water deficit conditions (*Abdelaal et al., 2017*). *Abdelaal et al. (2021a)*, *Abdelaal et al. (2021b)*, *Abdelaal et al. (2021c)*, *Abdelaal et al. (2021d)* and *Abdelaal et al. (2021e)* reported that yeast extract significantly increased garlic yield under drought conditions. Seed priming with *Bacillus thuringiensis* boosted adaptation in pea plants under drought conditions and increased the number of leaves, leaf area, and flowers per plant in both seasons (*Arafa et al., 2021*). Biochar treatments may play a significant role in the enhancement of morphological features under drought stress conditions since this treatment led to improved plant height and branch number of drought-stressed barley plants (*Abdelaal et al., 2022b*). In the experiments of *AlKahtani et al. (2021a)*, significant

decreases in morphological characteristics under water deficit stress conditions were shown. However, the application of salicylic acid led to the improvement of morphological characteristics and the overcoming of drought stress negative effects. Furthermore, the application of yeast extract caused a significant improvement in the morphological characteristics of maize plants under water deficit conditions (*Abdelaal et al., 2017*). In experiments with wheat, exogenously application of yeast extract and ascorbic acid led to the improve the plant height and leaves number under water deficit (*Abdelaal et al., 2021d*).

## Effect of drought stress on physiological and biochemical characteristic of plants

Antioxidant enzymes are a very important sign of drought stress conditions. In this context, *Abdelaal et al. (2021c)* reported that drought stress affects antioxidant enzyme activity, photosynthetic parameters, and relative water content in stressed garlic plants. Additionally, electrolyte leakage, and concentration of $H_2O_2$, and $O_{2-}$ were significantly higher in garlic plants under drought conditions (*Abdelaal et al., 2021c*). Furthermore, concentrations of chlorophyll a and chlorophyll b, and relative water content of pea plants significantly decreased under drought stress conditions (*Arafa et al., 2021*). On the other hand, $O_{2-}$ and $H_2O_2$ levels significantly increased in pea plants under drought stress. The maize plants exposed to water deficit stress showed that water deficit led to significant decreases in chlorophyll content in the stressed plants compared to control, whereas electrolyte leakage and MDA content significantly increased. Furthermore, a significant decrease in antioxidant compounds and protein was recorded in maize plants under water-deficit conditions (*Abdelaal et al., 2017*), while drought-stressed barley plants showed significant decreases in photosynthesis, relative water content, and water uptake (*Hafez et al., 2020a*; *Hafez et al., 2020b*).

## Effect of PGPB and safety compounds on physiological and biochemical characteristics of plants under drought conditions

Several studies were conducted to improve plants' growth and physiological features under water deficit stress conditions (Table 2). Yeast extract significantly improved antioxidant enzyme activity and relative water content and decreased ROS in garlic plants under drought conditions. Additionally, garlic's physiological and biochemical characteristics were improved with chitosan treatment under drought conditions, including relative water content and photosynthetic parameters (*Abdelaal et al., 2021c*). Furthermore, the number of seeds per pod, pod length, and dry weight of 100 seeds of pea plants were significantly increased with the application of carrot extract under drought stress conditions (*Arafa et al., 2021*). Also, *Bacillus thuringiensis* treatment improved the biochemical characteristics of pea plants such as chlorophyll a, chlorophyll b, relative water content, and decreased $O_{2-}$ and $H_2O_2$ under drought conditions. The carrot extract can be very significant due to the presence of essential vitamins such as vitamins A and C, which significantly improve plant growth and development by increasing plant height, leaf area, and number of leaves in pea plants. The positive impacts of carrot extract were also recorded on the physiological features of faba bean plants under drought conditions (*Kasim, Nessem & Gaber, 2019*).

The application of biochar led to an increase in chlorophyll a and b content, relative water content, and antioxidant compound activity in barley plants under drought-stress conditions (*Hafez et al., 2020a*; *Hafez et al., 2020b*). Under water deficit conditions, PGPB can be used to enhance the morphophysiological and biochemical characteristics *via* phosphorus solubilization, nitrogen uptake, and the availability of essential nutrients (*Kasim, Nessem & Gaber, 2019*; *Ghosh, Gupta & Mohapatra, 2019*). Treatment with PGPB such as *Serratia* spp. (*Heinrichs et al., 2009*), *Azoarcus* sp., *Pseudomonas* spp. (*Egener, Hurek & Reinhold-Hurek, 1999*; *Igiehon & Babalola, 2021*; *Sandhya et al., 2010*; *Zahir et al., 2009*), and *Rhizobium* spp. led to better morphophysiological features and yield of several plants (Table 2). Furthermore, the application of *Azospirillum* gave the best results regarding the drought tolerance of maize plants (*García et al., 2017*). The positive role of PGPB could be attributed to the improvement of water accessibility and antioxidant components as well as the production of osmoprotectants which improve plant growth under drought conditions. *Meenakshi et al. (2019)* stated that *Bacillus thuringiensis, Bacillus subtilis,* and *Bacillus megaterium* treatments significantly increased chlorophyll content in stressed plants. The useful effects of PGPB under stressful conditions might be due to the formation of osmolytes and the improvement of the antioxidant system.

# EFFECT OF SALT STRESS ON MORPHOLOGICAL, PHYSIOLOGICAL AND BIOCHEMICAL CHARACTERISTICS OF PLANTS

## Effect of salt stress on morphological characteristics

Under salinity conditions, plants display many variations and changes in all developmental stages. Salinity stress significantly decreases plant height, number of leaves, and leaf area in stressed plants such as calendula (*El-Shawa, Rashwan & Abdelaal, 2022*), sweet basil, and pea plants (*Jakovljević et al., 2017*; *Sapre, Gontia-Mishra & Tiwari, 2021*). Also, fresh weight and dry weight were significantly reduced under salinity conditions in barley (*Elsawy et al., 2022*), rice (*Mohamed et al., 2022*), basil (*Jakovljević et al., 2017*) and bean (*Kutlu & Gulmezoglu, 2023*). *Khedr et al. (2023)* reported that the salt-stressed wheat plants were negatively affected since most of the studied characteristics significantly decreased under salinity conditions. In the experiment with lettuce, both salinity concentrations (4 dS m$^{-1}$ and 8 dS m$^{-1}$) caused a significant decrease in leaf number and head weight in the stressed plants compared to the control in two seasons. This harmful effect of salt stress may be due to decreased water uptake from the soil, and the inhibition of cell elongation and cell division. Salinity stress is also followed by toxicities of Na+ and Cl−, which reduce the uptake of essential elements like nitrogen, phosphorus, and potassium (*ALKahtani et al., 2021b*) reducing the stressed plants' morphological features.

## Effect of PGPB and safety treatments on morphological characteristics of plants under salinity conditions

Many studies were conducted to find suitable methods to overcome the negative effects of salinity stress. *Bharti et al. (2016)* reported that the application of PGPB *Dietzia natronolimnaea* caused improvement in the morphological features of wheat plants

**Table 2  Treatments with safety compounds improving physiological and biochemical characteristics of economic plants under drought stress conditions.**

| Treatments | Improved physiological and biochemical characters | Plant species | References |
|---|---|---|---|
| PGPB | Act osmoregulatory | Some plants | *Soliman et al. (2018)* |
| Ascorbic acid (Vitamin C) | Protection of plant cells against toxic free radicals | Tomato | *Farooq et al. (2017)* |
| *Bacillus thuringiensis* | Physiological characters and seed yield | Pea | *Abdelaal (2015b)* |
| Yeast extract | Activity of antioxidant enzymes, photosynthetic process, and RWC | Garlic | *Abdelaal et al. (2021c)* |
| Mannitol | Super oxide and hydroxyl radical scavenging; Protection of chloroplasts against photo-oxidative stress | *Arabidopsis thaliana* | *Abebe et al. (2003)* |
| Melatonin | Concentration of photosynthetic pigments (chlorophyll a and b, and carotenoids) | Soybean | *Imran et al. (2023)* |
| *Bacillus cereus* | The significant increase in IAA, and zeatin | Walnut | *Liu et al. (2023)* |
| *Pseudomonas* spp. | Root elongation and NPK concentration | Soybean | *Igiehon & Babalola (2021)* |
| *Pseudomonas and Bacillus* | Physiological characters and yield | Tomato | *Nemeskéri et al. (2022)* |
| *Azotobacter* and *Pseudomonas* | Seed germination and seedlings' radical | *Spinacia oleracea* | *Petrillo et al. (2022)* |
| Proline | Protection of plant cells against the increased ROS production; water availability | Sugar beet, Bean and Soybean | *Abdelaal et al. (2021e)* and *Al-Shammari, Altamimi & Abdelaal (2023)* |
| Gamma-aminobutyric acid (GABA) | Cellular ion homeostasis | Arabidopsis | *Ghosh, Gupta & Mohapatra (2019)* |
| | Improve antioxidant enzymes | Tomato | |
| *Paenibacillus polymyxa* | Improve the physiological characteristics | Pepper | *Quyet-Tien et al. (2010)* |
| *Bacillus spp* and *Pseudomonas spp* | Increased fruit yield | Pepper | *Admassie et al. (2022)* |
| *Bacillus* spp. | Improving the physio-chemical properties | Mung bean | *Kumar et al. (2023)* |

under salt-stress conditions. *Bacillus thuringiensis* treatment increased leaves number, head weight, and total yield in lettuce plants under salinity stress compared to untreated plants (*AlKahtani et al., 2021a*). In the study of *El-Flaah et al. (2021)*, the treatment with Rhizobium improved the growth characteristics of salt-stressed faba bean plants such as branch number, plant height, and flower number. Additionally, *Abdelaal, Mazrou & Hafez (2020)* reported that stress tolerance-inducing compounds alleviate salinity stress effects on sweet pepper plants and improve growth characteristics. The application of chitosan caused a significant improvement in morphological characteristics of sweet pepper plants including plant height, leaf number, and fresh shoot weight under salt stress (*Abdelaal et al., 2021c*). *Al-Shammari, Altamimi & Abdelaal (2023)* found an improvement in the seed yield of pea plants under salinity conditions with melatonin treatment, the growth characteristics also were improved under these conditions. One of the crucial treatments in alleviating the negative effects of salinity stress is biochar, which may overcome the harmful effects and improve the growth characteristics of faba bean plants (*El Nahhas et al., 2021*).

## Effect of salt stress on physiological and biochemical characteristics of plants

Salinity stress significantly affects the physiological and biochemical characteristics of plants, resulting in an imbalance of ions and a decrease in nutrient uptake, especially nitrogen, phosphorus, and potassium (*Kumar et al., 2017*; *Putra, Santosa & Salsinha, 2023*). The harmful effects on physiological and biochemical features may be due to the decrease in the synthesis of growth-promoting hormones. IAA is a hormone that stimulates cell division and elongation (*Putra, Santosa & Salsinha, 2023*). High salinity levels can decrease amino acid production (like tryptophan) which is involved in the IAA synthesis, resulting in decreased IAA concentration. Many physiological and biochemical changes were observed under salinity conditions, such as the synthesis of compatible solutes, antioxidant components, and stress proteins, as well as the excessive production of ROS, which negatively affects cellular structures, inhibits carbon fixation, decreases the photosynthetic rate, and causes hormonal balance disorder (*Abdelaal, Mazrou & Hafez, 2020*). Moreover, electrolyte leakage, MDA content, and superoxide and hydrogen peroxide production increase, while chlorophyll content and relative water content decrease. The synthesis of antioxidant compounds, including both enzymatic and non-enzymatic antioxidants, is one of the mechanisms to deal with salinity and it is based on the protection of cells against oxidative damage. Glutathione reductase, peroxidase, catalase, superoxide dismutase, and ascorbate peroxidase are the main enzymatic antioxidants (*Usman et al., 2023*; *Jahanbani et al., 2023*), while non-enzymatic antioxidants including ascorbic acid, proline, carotenoids and $\alpha$-Tocopherols (vitamin E) (*Rigi, Saberi & Ebrahimi, 2023*; *Hafez et al., 2020a*; *Hafez et al., 2020b*). *El Nahhas et al. (2021)* reported that salinity stress significantly decreased relative water content and chlorophyll concentration, while lipid peroxidation, electrolyte leakage, and ROS significantly increased in salt-stressed faba bean plants. Additionally, salt concentration was positively correlated with the content of secondary metabolites such as phenolics (including flavonoids), and proline (*Beheshti, Khorasaninejad & Hemmati, 2023*). The decrease in the content of photosynthetic pigments under salinity conditions could be due to the chlorophyll photooxidation and chloroplasts oxidative damage under salinity conditions (*Beheshti, Khorasaninejad & Hemmati, 2023*; *Singh et al., 2022*).

## Mitigation of salt stress effects on the physiological and biochemical characteristics of plants

It can be concluded that the adverse effects of salinity can be detected in all stages of plant growth and development, including both vegetative and reproductive stages in numerous plants (*El-Flaah et al., 2021*; *Aziz et al., 2023*). Application of PGPB was used to improve plant growth and increase productivity under salinity stress (Table 3). It is shown that plant growth-promoting rhizobacteria can improve physiological and biochemical characteristics such as water absorption, nutrient uptake, and photosynthesis, and promote plant growth and development. PGPB can produce essential compounds such as siderophores and ACC-deaminase which increase the availability of some nutrients in the soil. For example, the *Enterobacter* treatment led to increased ACC deaminase activity and alleviated salinity stress by reducing ethylene concentration (*Singh et al., 2022*; *Agha et al., 2023*). Under

**Table 3  Mitigation of salinity stress effects on the physiological and biochemical characters of economic plants.**

| Treatments | Improved physiological and biochemical characters under salinity stress | Plant species | References |
|---|---|---|---|
| *Enterobacter cloacae* | Production of ACC deaminase | Wheat | *Singh et al. (2022)* |
| *Bradyrhizobium japonicum* + *Enterobacter Delta PSK* | Reduction of MDA, electrolyte leakage and hydrogen peroxide | Soybean | *Agha et al. (2023)* |
| *Piriformospora indica* | Alleviation the oxidative damages | *Trigonella* | *Bisht et al. (2022)* |
| *Enterobacter cloacae* PM23 | Regulation of antioxidant defense, and solutes accumulation | Maize | *Ali et al. (2022)* |
| *Enterobacter cloacae* and *Bacillus drentensis* | water uptake and nutrient accessibility | Mung bean | *Mahmood et al. (2016)* |
| *Halomonas* sp. | Siderophores production | Rice | *Mukherjee, Mitra & Roy (2019)* |
| *Pesudomonas* | Exopolysaccharides secretion | Rice | *Sen & Chandrasekhar (2014)* |
| *Pseudomonas* sp. | ACC-deaminase production | Wheat | *Choudhary et al. (2022)* |
| *Achromobacter* sp. and *Bacillus* sp. | ACC deaminase and siderophore production | Rice | *Zahir et al. (2009)* |
| *Klebsiella* sp. | Production of IAA, antioxidant enzymes and proline | *Avena sativa* | *Sapre, Gontia-Mishra & Tiwari (2018)* |
| *Pseudomonas azotoformans FAP5* | Both morphological and physiological attributes | Wheat | *Ansari, Jabeen & Ahmad (2021)* |
| *Gluconacetobacter diazotrophicus* | chlorophyll content | Maize | *Tufail et al. (2021)* |
| *Azospirillum* sp. | phosphate solubilization | Chickpea | *Patel et al. (2021)* |
| *Pseudomonas pseudoalcaligenes* | antioxidant enzymes activity | Soybean | *Yasmin et al. (2020)* |
| Serotonin | Decrease MDA contents and electrolyte leakage | Tomato | *Akcay & Okudan (2023)* |
| Melatonin | Increase the activities of SOD, POD, and CAT | Alfalfa | *Guo et al. (2023)* |
| Yeast extract + proline | Improved chlorophyll contents, decrease the levels of super oxide and electrolyte leakage | *Solidago virgaurea* | *Arnao & Hernández-Ruiz (2017)* |

salt stress conditions, MDA production, electrolyte leakage, and concentration of $H_2O_2$ significantly decreased with the application of *Bradyrhizobium japonicum* + Enterobacter in the stressed soybean plants, while chlorophyll concentrations increased (*Bisht et al., 2022*). *AlKahtani et al. (2021a)* and *Bisht et al. (2022)* reported that the application of PGPB markedly improved photosynthetic rate, stomatal conductance, and chlorophyll and carotenoid content in lettuce and *Trigonella* under salinity stress. Moreover, *Ali et al. (2022)* found that inoculation with *Enterobacter cloacae* improved relative water content, flavonoid, and protein content in maize under salt stress (*Mahmood et al., 2016*). Also, *Mukherjee, Mitra & Roy (2019)* reported that the application of *Halomonas* sp. promotes the growth characteristics of rice plants and increases siderophore production under salinity stress. Likewise, *Klebsiella variicola* produced many compounds under salinity conditions such as siderophore, 1 aminocyclopropane-1-carboxylate deaminase, IAA, exopolysaccharides, and antioxidant enzymes (*Sen & Chandrasekhar, 2014*).

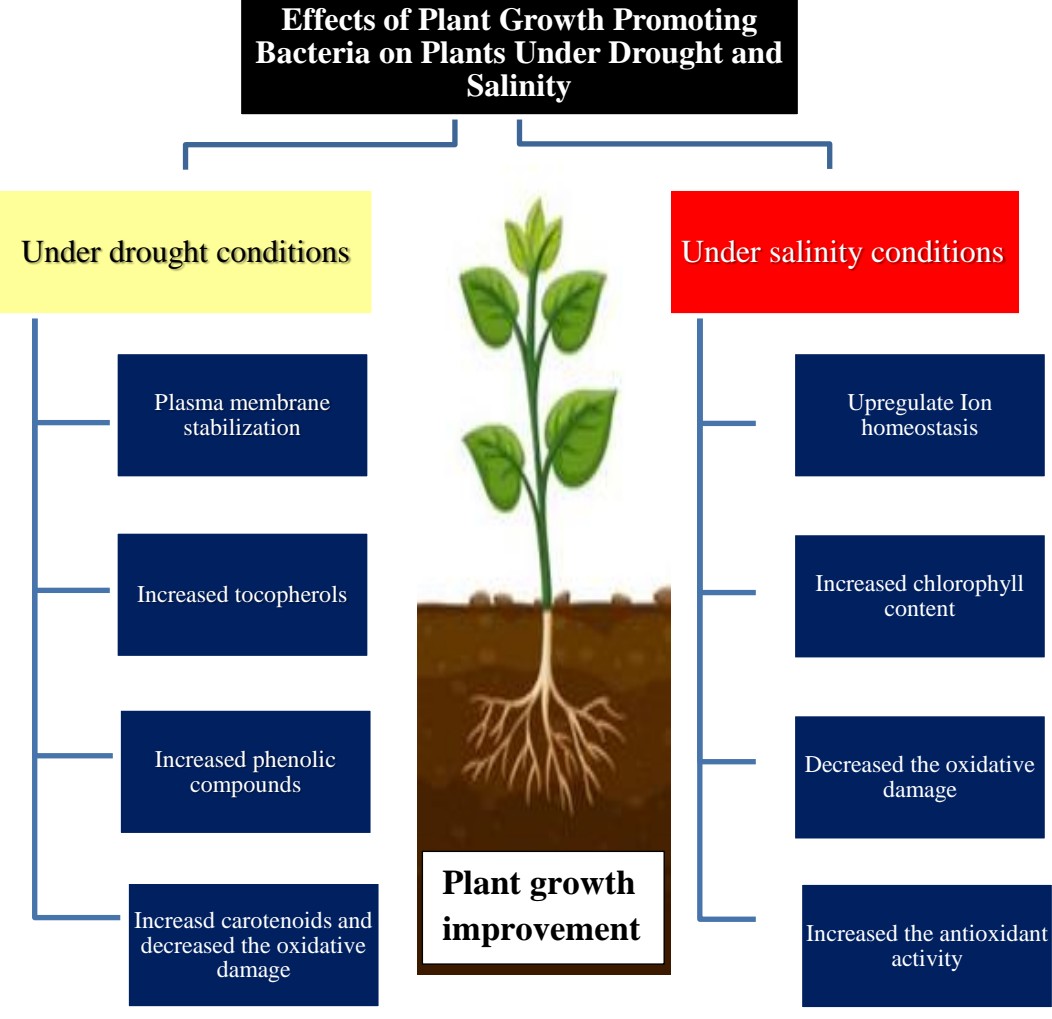

**Figure 1** Effect of plant growth promoting bacteria on plant growth under drought and salinity stress conditions.

The positive role of PGPR may be due to improved uptake of nitrogen, phosphorus, and potassium (*Zahir et al., 2009*; *Choudhary et al., 2022*). PGPR may trigger the antioxidant defense system to scavenge ROS which are produced as indicators of salt stress in many plants (*Sapre, Gontia-Mishra & Tiwari, 2018*; *Ansari, Jabeen & Ahmad, 2021*). Previous studies showed that the application of *Pseudomonas* sp. and *Serratia* sp. led to improved physiological characteristics in the salt-stressed wheat plants (Table 3) *via* enhanced ACC-deaminase production in plants (*Tufail et al., 2021*; *Patel et al., 2021*). Also, the secretion of phytohormones and nitrogen fixation is one of the various mechanisms to decrease the negative effect of salinity stress. Generally, PGPB can play an important role in improving the plant tolerance to drought and salinity stress conditions (Fig. 1).

Furthermore, the application of melatonin caused significant increases in chlorophyll concentrations and relative water content in the stressed pea plants. However, a significant

decrease in the MDA production, EL%, and $H_2O_2$ and $O_2-$ concentration was recorded in the stressed plants compared with the stressed pea plants without treatments (*Kour et al., 2020*). Serotonin is a vital bioregulator that is formed during the biosynthesis pathway of melatonin and has light-absorbing properties. Serotonin and melatonin are indolamines, they have a pivotal role for life in living organisms (*Azmitia, 2011*). Serotonin application led to decreased MDA content and EL% level of tomato plants, while relative water content was increased under salinity and drought stress conditions (*Akcay & Okudan, 2023*).

Exogenously applied melatonin mitigates the negative effect of salt stress by improving the IAA content, photosynthesis rate, and photosynthesis efficiency and reducing the accumulation of $H_2O_2$ in stressed wheat seedlings (*Ke et al., 2018*). *Guo et al. (2023)* reported that melatonin applied as foliar spray increased root length, leaf area, proline content, and soluble protein as well as the activity of antioxidant enzymes such as catalase and peroxidase in the salt-stressed alfalfa plants, while the levels of MDA were decreased. Melatonin can also mitigate the adverse effects of salinity and drought (*Wang, Reiter & Chan, 2017*). Furthermore, melatonin stimulates the essential enzymes, such as CAT, APX, SOD, POD, and GR, as well as some important metabolites such as flavonoids, glutathione, ascorbate, and anthocyanins which activate the antioxidant system and protect plants from oxidative damage under various stresses. Melatonin can also play a significant role in controlling $H_2O_2$ levels (*Anwar et al., 2023*). The positive effect of melatonin may be due to its role in maintaining root development and photosynthetic rate under salinity conditions (*Arnao & Hernández-Ruiz, 2017*). Application of yeast extract and proline led to increased concentration of elements such as N, P, K, and Ca in calendula plants under salinity stress, whereas concentration of Na, proline, $O_2-$ and $H_2O_2$, and electrolyte leakage were decreased in the stressed plants (*El-Shawa, Rashwan & Abdelaal, 2022*).

The application of yeast extract and/or gibberellic acid ($GA_3$) gave the highest increase in plant growth characteristics such as plant height and fresh and dry weight of shoots, as well as photosynthetic efficiency, macronutrient content, total soluble sugars, and total phenolics in *Solidago* plants under alkaline soil conditions (*Youssef et al., 2022*). Additionally, the application of biochar and compost (alone or together) led to a significant improvement in plant growth compared to the control. Furthermore, the combined application of compost and biochar significantly reduced the sodium in the shoots and roots of tomato plants (*Ud Din et al., 2023*). It was also observed that the application of arbuscular mycorrhizal fungi (*Rhizophagus intraradices*) led to an increase in nutrient uptake, accumulation of compatible osmolytes, and decreased electrolyte leakage in *Pisum sativum* under salinity stress conditions (*Parihar et al., 2020*). *Shahzad et al. (2017)* found that the ability of rice plants to tolerate salinity stress was significantly increased by inoculation with arbuscular mycorrhizal fungi, which alter the endogenous phytohormone and essential amino acids such as glutamic acid, phenylalanine, cysteine, and proline. Additionally, arbuscular mycorrhizal fungi can improve the availability of nutritional elements *via* biological processes (*Ma, Vosátka & Freitas, 2019*) under various environmental factors. *Navarro-Torre et al. (2023)* reported that the application of halotolerant/halophilic bacteria can be one of the important strategies in improving plant tolerance to salinity stress in grapevines in the context of climate change. Furthermore, PGPB can improve plant tolerance to

several abiotic stresses, especially salinity, by regulating the enzymatic and nonenzymatic antioxidants to prevent the accumulation of ROS, which results in improved plant growth (*Abdou et al., 2023*).

## EFFECT OF SALINITY AND DROUGHT STRESSES ON ANATOMICAL CHARACTERISTICS OF PLANTS

Plants display many changes in anatomical structures under salinity and drought conditions such as a decrease in the thickness of lamina, midvein, palisade and spongy leaf tissues (Fig. 2). Additionally, salinity and drought can cause detrimental effects on the anatomical structure of plant stems. *Abdo et al. (2012)* reported that salinity stress decreased the thickness of the soybean leaves' lamina, spongy, and palisade tissues. *El-Banna & Abdelaal (2018)* found that high NaCl levels induced ultrastructural changes in the mesophyll tissue of strawberry leaflets such as the disintegration of grana staking, swelling of thylakoids (Fig. 3), increased number of starch grains and plastoglobuli, as well as increased size of mitochondria. Also, salinity led to the shrinkage of plasma membranes, increased vesicle formation, and increased cell wall thickness (*El-Banna & Abdelaal, 2018*). However, the application of yeast extract and garlic extract led to an improvement of the anatomical characteristics of soybean plants under stress conditions (*Abdo et al., 2012*). The positive effect of yeast extract and garlic extract on the anatomical characteristics of soybean plants could be related to the improvement in plant growth. In the experiment of faba bean, *Taie et al. (2013)* reported that salinity stress led to a decrease in the thickness of the epidermis, cortex, phloem tissue, xylem tissue, as well as parenchyma tissue. Also, vessel diameter and internode diameter were decreased due to reduced stem wall thickness and hollow pith diameter reduction. Under salinity conditions, the midvein and lamina thickness of the leaflet was decreased due to the decrease in thickness of both palisade and spongy tissues (*Taie et al., 2013*).

It is shown that salinity at 6,000 ppm reduced the midvein's thickness, vascular bundle length and width, xylem rows, and vessel diameter in cowpea leaflets. Also, the stem diameter, cortex thickness, and thickness of phloem and xylem tissues decreased in the stressed cowpea plants compared with the control (*El-Taher et al., 2022*). *Taha et al. (2021)* found that the application of yeast extract led to an improvement in the anatomical features of the lupine stems such as stem diameter, cortex thickness, xylem vessel diameter, and number of pith layers in the salt-stressed lupine plants. The damaging effects of drought can be seen in drought-stressed eggplant through the changes in lamina thickness, length and width of the vascular bundle, and the midvein thickness. However, the application of 100 ppm ZnO NP reduced the damaging effect of drought on the stressed eggplant (*Semida et al., 2021*). Likewise, a significant reduction in anatomical characteristics of leaves such as lamina thickness, palisade and spongy tissue thickness, vascular bundle width, and a number of xylem vessels were recorded in the drought-stressed fenugreek plants (*Abou-Sreea et al., 2022*). Under drought stress conditions, negative effects were recorded on the ultrastructure of sugarcane leaves such as deformation of chloroplasts, plasmolysis of mesophyll tissue, alteration of lamellae in the chloroplast, also, chloroplasts getting round,

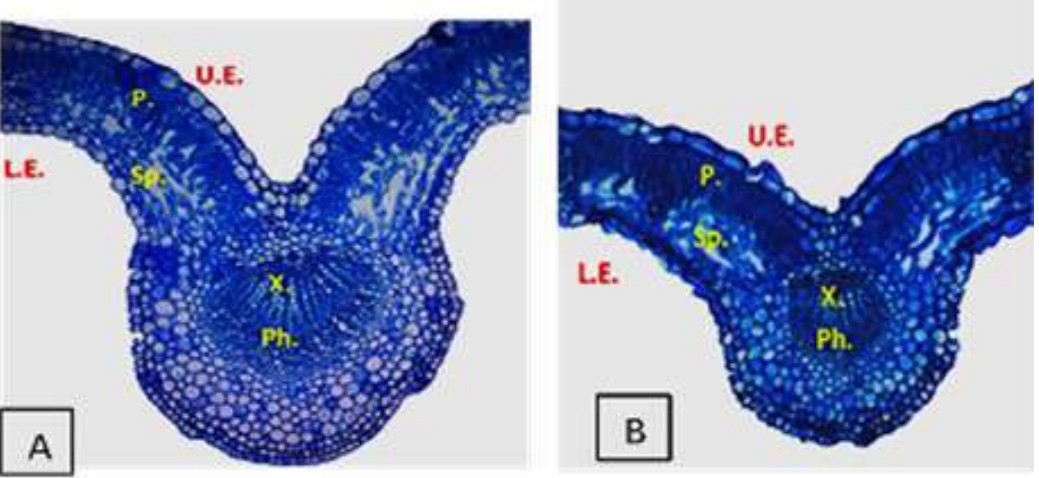

**Figure 2** **Cross sections micrographs of strawberry leaflet under salinity conditions.** A (control), B (68 mM NaCl) (*El-Banna & Abdelaal, 2018*). U.E., upper epidermis; P., palisade tissue; Sp., spongy tissue; X., xylem; Ph., phloem; L.E., lower epidermis. (X100).

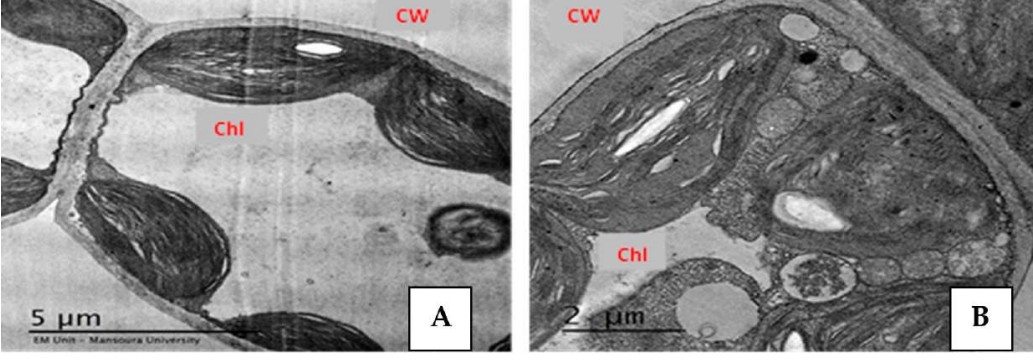

**Figure 3** **Effect of salinity stress on strawberry leaflet showing alterations in the ultrastructure of leaflet organelles.** A (control), B (68 mM NaCl) (*El-Banna & Abdelaal, 2018*). Ch, chloroplast; CW, cell wall. Scale bar = (A) 5 μm and (B) 2 μm.

swollen, and produced osmiophilic particles (*Zhang et al., 2015*). The negative effects of drought on anatomical features could be attributed to the reduction of water supply and nutrient uptake from the soil, which, negatively affect the anatomical structure of plants.

## CONCLUSIONS AND FUTURE PERSPECTIVES

The issue of abiotic stress is very significant in the agricultural sectors, where plants must deal with salinity and drought conditions which cause a sharp decrease in crop production. It can be seen that treating plants with PGPB and safety treatments has multiple mechanisms for improving growth and yield production. The adverse effects of drought and salinity can be mitigated by applying PGPB, arbuscular mycorrhizal

fungi, yeast extract, melatonin, biochar, chitosan, and proline. These safety treatments can improve morphoanatomical attributes such as the diameter of the xylem vessel, pith layer, cortex thickness, and overall stem diameter. They also regulate ion homeostasis and osmolyte accumulation, enzymatic and non-enzymatic antioxidant activities, and play a pivotal role in phytohormone biosynthesis. Many studies showed that the application of safety treatments significantly improved the physiological and biochemical attributes in economically important plants under drought and salinity stress conditions, such as chlorophyll concentrations, carotenoids content, relative water content, osmoprotectants accumulation, organic compounds production, as well as scavenge of the ROS. It can be concluded that safety and eco-friendly treatments may have a significant role in developing economically important plants with improved tolerance to environmental stresses. Still, further studies are required to investigate the molecular mechanisms employed in order to thoroughly understand and further apply species-specific biofertilizers and increase the tolerance to salinity and drought.

### Funding

This work was supported by the deanship at the University of Ha'il, Saudi Arabia, through project number RG-23 100. The funders had no role in study design, data collection and analysis, decision to publish, or preparation of the manuscript.

### Grant Disclosures

The following grant information was disclosed by the authors:
University of Ha'il, Saudi Arabia: RG-23 100.

### Competing Interests

Diaa Abd El Moneim is an Academic Editor for PeerJ.

### Author Contributions

- Wasimah Buraykan Alshammari conceived and designed the experiments, performed the experiments, analyzed the data, prepared figures and/or tables, authored or reviewed drafts of the article, and approved the final draft.
- Kholoud Alshammery conceived and designed the experiments, performed the experiments, analyzed the data, prepared figures and/or tables, authored or reviewed drafts of the article, and approved the final draft.
- Salwa Lotfi conceived and designed the experiments, performed the experiments, analyzed the data, prepared figures and/or tables, authored or reviewed drafts of the article, and approved the final draft.
- Haya Altamimi conceived and designed the experiments, performed the experiments, analyzed the data, prepared figures and/or tables, authored or reviewed drafts of the article, and approved the final draft.

- Abeer Alshammari conceived and designed the experiments, performed the experiments, analyzed the data, prepared figures and/or tables, authored or reviewed drafts of the article, and approved the final draft.
- Nadi Awad Al-Harbi conceived and designed the experiments, performed the experiments, analyzed the data, prepared figures and/or tables, authored or reviewed drafts of the article, and approved the final draft.
- Dragana Jakovljević conceived and designed the experiments, performed the experiments, analyzed the data, prepared figures and/or tables, authored or reviewed drafts of the article, and approved the final draft.
- Mona Hajed Alharbi conceived and designed the experiments, performed the experiments, analyzed the data, prepared figures and/or tables, authored or reviewed drafts of the article, and approved the final draft.
- Moustapha Eid Moustapha conceived and designed the experiments, performed the experiments, analyzed the data, prepared figures and/or tables, authored or reviewed drafts of the article, and approved the final draft.
- Diaa Abd El-Moneim conceived and designed the experiments, performed the experiments, analyzed the data, prepared figures and/or tables, authored or reviewed drafts of the article, and approved the final draft.
- Khaled Abdelaal conceived and designed the experiments, performed the experiments, analyzed the data, prepared figures and/or tables, authored or reviewed drafts of the article, and approved the final draft.

## Data Availability

This is a literature review.

## Supplemental Information

Supplemental information for this article can be found online at http://dx.doi.org/10.7717/peerj.17286#supplemental-information.

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
