# Peer review of "Improvement of morphophysiological and anatomical attributes of plants under abiotic stress conditions using plant growth-promoting bacteria and safety treatments"

_PeerJ, doi:10.7717/peerj.17286_

## Round 0.1 · original submission · Minor Revisions

· Academic Editor

Minor Revisions

Although one reviewer requested a major revision, my opinion is that the corrections they requested were minor changes. Be sure to make your changes appropriately and submit the revised manuscript at the specified time.

**Language Note:** The review process has identified that the English language must be improved. PeerJ can provide language editing services - please contact us at [email protected] for pricing (be sure to provide your manuscript number and title). Alternatively, you should make your own arrangements to improve the language quality and provide details in your response letter. – PeerJ Staff

Reviewer 1 ·

Basic reporting

The idea of this review is very interesting, it covers two major abiotic stresses and the responses of many plants to these stresses. Also the review explain the role of plant growth promoting bacteria to mitigate the harmful effects of these stresses on morphological, physiological and anatomical characters. The Introduction is adequate and introduce the subject and make it clear to the audience.

Experimental design

Review content is within the Aims and Scope of the journal the review organized logically into coherent paragraphs/subsections.

Validity of the findings

No comment

Additional comments

There are minor issues in the pdf version, the authors should follow the comments in the attached pdf version

Annotated reviews are not available for download in order to protect the identity of reviewers who chose to remain anonymous.

Reviewer 2 ·

Basic reporting

All the critics, comments and recommendations are avaliable in the manuscript.

Experimental design

no comment. its is review article.

Validity of the findings

no comment. it is review article.

Additional comments

This review article is written on the harms of two major abiotic stress factors: drought and salinity. Indeed, it focuses on these. In the scientific world, there are countless articles demonstrating the damages caused by these two major abiotic stress factors. I believe the time has come to shift towards researching effective methods of combating these stress factors and to present solutions. When everyone points out the darkness, it becomes clear that it's time to offer solutions to escape this darkness. Although this article presents some solutions, they are insufficient, and there are many methods to overcome drought, for example, water harvesting, developing varieties resistant to drought stress, etc. The problem of salinity is actually a cause of drought. Even when there is water in the environment, the plant cannot absorb it and ends up drying out.

Annotated reviews are not available for download in order to protect the identity of reviewers who chose to remain anonymous.

Reviewer 3 ·

Basic reporting

In the review titled (Improvement of anatomical and morphophysiological attributes of economic plants under abiotic stress conditions using plant growth promoting bacteria and safety treatments), authors summarize the pivotal role of various PGPB and eco-friendly treatments as a promising strategy to alleviate the negative impacts of drought and salinity and to increase the yield and production of economically important plants under stressful conditions.
The review is interesting and is generally well-written and structured. The survey methodology was successful, and the data was well understood and modeled in detail.
In addition, the text contains relevant paragraphs that have been discussed. The selection of the bibliography is appropriate to the content of the review. Some minor errors appeared throughout the Manuscript.
- Authors should scan the review for minor punctuation and English errors. (see attached file)
- The title needs to be modified.
- Arrange the keywords in alphabetical order.
- The introduction is appropriate but needs further improvements, especially the study hypothesis, which should be added for the last five years. What are the expected outcomes? Also, Provide a statement of novelty.
- Conclusion: Improve this part concerning formulated objectives.
- Cross-check the references in the text and reference cite.

Experimental design

Its fine.

Validity of the findings

1- Provide specific quantitative data points or ranges where applicable to illustrate the extent of improvement observed with each safety treatment. For example, instead of stating that treatments "significantly improved physiological and biochemical attributes," specify the percentage increase or decrease in key parameters such as chlorophyll concentrations or relative water content.

2-Acknowledge potential limitations of the studies reviewed or conducted, such as sample size, experimental design, or variability in environmental conditions. Addressing limitations helps to contextualize the findings and demonstrates a critical appraisal of the research.

3-Highlight areas where further research is needed to address gaps in understanding. For instance, identify specific aspects of molecular mechanisms that require investigation to fully elucidate the mode of action of safety treatments in improving plant tolerance to abiotic stress.

Annotated reviews are not available for download in order to protect the identity of reviewers who chose to remain anonymous.

---

## Round 0.2 · accepted · Accept

· Academic Editor

Accept

Dear authors,

Your manuscript was highly improved after the revisions that were made according to the reviewer and editorial comments. Congratulations

Reviewer 1 ·

Basic reporting

No comment

Experimental design

No comment

Validity of the findings

No comment

Additional comments

No comment